# Brain Endothelial Cells Play a Central Role in the Development of Enlarged Perivascular Spaces in the Metabolic Syndrome

**DOI:** 10.3390/medicina59061124

**Published:** 2023-06-11

**Authors:** Melvin R. Hayden

**Affiliations:** Department of Internal Medicine, Endocrinology Diabetes and Metabolism, Diabetes and Cardiovascular Disease Center, University of Missouri School of Medicine, One Hospital Drive, Columbia, MO 65211, USA; mrh29pete@gmail.com; Tel.: +1-573-346-3019

**Keywords:** brain capillary endothelial cell(s)

## Abstract

Brain capillary endothelial cell(s) (BECs) have numerous functions, including their semipermeable interface-barrier (transfer and diffusion of solutes), trophic (metabolic homeostasis), tonic (vascular hemodynamics), and trafficking (vascular permeability, coagulation, and leukocyte extravasation) functions to provide brain homeostasis. BECs also serve as the brain’s sentinel cell of the innate immune system and are capable of antigen presentation. In metabolic syndrome (MetS), there are two regions resulting in the proinflammatory signaling of BECs, namely visceral adipose tissue depots supplying excessive peripheral cytokines/chemokines (***p***CCs) and gut microbiota dysbiotic regions supplying excessive soluble lipopolysaccharide (sLPS), small LPS-enriched extracellular vesicle exosomes (lpsEVexos), and ***p***CCs. This dual signaling of BECs at their receptor sites results in BEC activation and dysfunction (BEC*act/dys*) and neuroinflammation. sLPS and lpsEVexos signal BECs’ toll-like receptor 4, which then signals translocated nuclear factor kappa B (NFkB). Translocated NFkB promotes the synthesis and secretion of BEC proinflammatory cytokines and chemokines. Specifically, the chemokine CCL5 (RANTES) is capable of attracting microglia cells to BECs. BEC neuroinflammation activates perivascular space(s) (PVS) resident macrophages. Excessive phagocytosis by reactive resident PVS macrophages results in a stagnation-like obstruction, which along with increased capillary permeability due to BEC*act/dys* could expand the fluid volume within the PVS to result in enlarged PVS (EPVS). Importantly, this remodeling may result in pre- and post-capillary EPVS that would contribute to their identification on T2-weighted MRI, which are considered to be biomarkers for cerebral small vessel disease.

## 1. Introduction

It is important to initially illustrate the origins of the fluid-filled perivascular spaces (PVS) and their relationship to the different capillary types, including precapillary arterioles, true capillaries, and post-capillary venules. The surface pia arteries and arterioles reside within the subarachnoid (SAS). As pial arteries and precapillary arterioles penetrate the cortical grey and subcortical white matter, they carry with them the lining of pia matter that become adherent to the basement membrane(s) (BMs) of the adjacent astrocyte end-feet BM forming the glia limitans abluminal boundary of the PVS. The luminal boundary of the PVS is the BM that is shared between BECs and pericytes, and these luminal and abluminal boundaries form the fluid-filled spaces referred to as the Virchow-Robin spaces or PVS. PVS serve as conduits for the delivery of cerebrospinal fluid (CSF) from the subarachnoid space (SAS) to the parenchyma via the precapillary arterioles. The post-capillary venule PVSs serve as conduits for the removal of the admixture of CSF and interstitial fluid (ISF) and metabolic waste to the SAS CSF and arachnoid granulations to the dural venous sinuses (Figure 1) [1]. 

Inflammation is a major cause of blood-brain barrier (BBB) dysfunction and disruption [2,3,4,5]. Further, disruption of BBB is known to result in increased permeability with leakage of plasma and neurotoxins into the brain interstitium and the PVS. Importantly, neurovascular unit (NVU) BBB disruption is found in multiple clinical disease states such as sepsis, HIV encephalopathy, obesity, metabolic syndrome (MetS), prediabetes, manifest T2DM, late-onset Alzheimer’s disease (LOAD), Parkinson’s disease (PD), multiple sclerosis (MS), and many others [2,3,4,5,6]. The administration of lipopolysaccharide (LPS) by intraperitoneal or intravenous injections is known to create classic models for the study of inflammation-induced BBB dysfunction and disruption of the NVU and neuroinflammation [6,7,8]. LPS, also known as an endotoxin and a pathogen-associated molecular pattern(s) (PAMP), has been studied regarding its mechanism of action on the mammalian immune system for at least 150 years [9]. LPS is a glycolipid (Lipid A) polysaccharide derived from Gram-negative bacterial plasma membranes that result in inflammation and neuroinflammation [9]. LPS may be considered to be a byproduct of both the gut dysbiosis microbiota and infections and is known to be increased in obesity, MetS, and T2DM [10,11,12]. Besides affecting brain endothelial cell(s) (BECs), LPS is also known to alter the other cells of the NVU [6,7,8]. The effects of LPS on the BBB can be direct, with LPS directly binding to BECs, or indirect, with LPS inducing the release of substances from cells, including BECs, pericyte(s) (Pcs), and astrocyte(s) (Acs), which then affect the BBB [6,7,8]. The direct effects of LPS initially involve peripherally derived LPS binding to blood-borne lipopolysaccharide-binding protein (LBP) that also incorporates a soluble cluster of differentiation 14 (CD14). This LPS/Lipopolysaccharide binding protein (LBP)/CD14 cluster binds to toll-like receptor 4 (TLR4) along with myeloid differentiation factor-2 (MD2) that initiates TLR4 proinflammatory signaling on BECs (Figure 2) [5,6,8].

Additionally, visceral adipose tissue (VAT) and gut microbiota dysbiosis-derived peripheral cytokines/chemokines (***p***CCs) are capable of being transported across the BBB. Indeed, several cytokines have been shown to cross the BBB by way of saturable transport systems without the disruption of BBB tight and adherens junctions, which also allow for the paracellular entry of *p*CCs (Figure 2) [6]. 

TLR4, a transmembrane protein and a member of the toll-like receptor family, belongs to the pattern recognition receptor family [9]. LPS-injected models are classic models for studying inflammation/neuroinflammation-induced BBB disruption and increased permeability [5,6,8]. The indirect effects of LPS are equally important in causing neuroinflammation and possibly EPVS; however, they were not specifically pointed out in Figure 2. Indirect effects of soluble LPS (sLPS) or LPS extracellular vesicle exosomes (lpsEVexos) may be mediated on peripheral cells, including circulating immune cells (leukocytes), which are capable of generating large amounts of ***p***CCs. In addition to metainflammation, ***p***CCs generated by the VAT depots and the gut microbiota dysbiosis interact with the luminal surface of BECs and alter BBB functions, including increased BBB permeability to immune cells [6]. The blood-borne peripherally-derived sLPS and lpsEVexos have been demonstrated to penetrate the BBB (even disrupted BBBs) only minimally [6,13]. Therefore, the effects of LPS on NVU functions are presumed to be by both direct and indirect mechanisms, with indirect mechanisms possibly playing the more predominant role via TLR4 signaling, as depicted in Figure 2 [6,13,14].

The MetS may be characterized as a cluster of multiple interconnected risk factors and variables that are associated with an increased risk for the development of atherosclerotic cerebrocardiovascular disease (CCVD) and T2DM [15,16,17]. Additionally, visceral obesity and MetS are associated with an increased risk for the development of EPVS [18,19,20]. Further, it is specifically known that IR (a central feature of MetS) along with its core risk components hyperlipidemia, hyperinsulinemia, hypertension (HTN), and hyperglycemia with or without T2DM is a risk factor for the development of EPVS (Figure 3) [20].

There are four core features (the four arms of the letter X), which include hyperlipidemia, hyperinsulinemia, hypertension, and hyperglycemia [15,16,17]. In addition to the MetS, EPVS and SVD are also known to be associated with aging [15,21,22,23]. Obesity and the MetS are increasing globally due to an aging population, urbanization, sedentary lifestyles, and increased caloric diets high in fat, sucrose, fructose, and glucose [15,16,17]. Currently, our global population is now considered to be one of the oldest attained in our history [24]. Therefore, it is not surprising that we are observing a global increase not only in the MetS but also an increase in EPVS and SVD [21,22,25,26]. Recently, it has been demonstrated that capillary rarefaction is associated with and accentuated when there is decreased NO bioavailability, as in the MetS (Figure 3) [27,28]. Capillary rarefaction (loss of capillaries) is a condition wherein there is a decrease in small vessel capillary density that occurs in the brain. This decrease in the number of capillaries may have regional variations with certain disease processes and vary between different organs [27,28]. Visceral obesity, MetS, decreased bioavailable NO, and advancing age contribute to cerebromicrovascular—capillary rarefaction (Figure 3) [15,19,27,28,29]. Recently, it has been hypothesized that capillary rarefaction leaves an empty space within the PVS that is subsequently filled with a mixture of interstitial waste fluid and may give rise to an increase in total fluid within the PVS, which may result in EPVS [1]. Given that this mechanistic hypothesis is possible, there will still be more research necessary to support this hypothesis. Accumulation of VAT, IR, leptin resistance, oxidative stress, neuroinflammation, elevated lipids, and/or glucose contributes to ultrastructure remodeling in the obese, IR, diabetic *db/db*, and BTBR *ob/ob* preclinical mice models, which contribute to neuroinflammation, accelerated brain aging, neurodegeneration, brain atrophy, and impaired cognition [1,15,30,31]. Interestingly, each of the four arms of the MetS—hyperlipidemia, hyperinsulinemia of insulin resistance, hypertension, and hyperglycemia—serve either directly or indirectly as neurotoxins and are associated with an increased risk for the development of EPVS (Figure 2) [20]. 

In the obese, insulin-resistant, diabetic female *db/db* preclinical mouse models, the descending thoracic aorta contains a markedly expanded perivascular adipose tissue (PVAT) depot. In these depots, the normal brown adipose tissue (BAT) has differentiated to hypertrophic unilocular white adipocyte tissue (WAT) with extremely thinned plasma membranes that are prone to rupture and expel their toxic proinflammatory lipid contents including extracellular vesicle exosomes (EVexos). These expelled toxic proinflammatory lipids, including long-chain saturated free fatty acids and EVexos, are capable of inciting an accumulation of reactive macrophages with the creation of crown-like structure macrophages resulting in PVAT inflammation [32,33]. Recently, the author has been able to demonstrate crown-like structures utilizing a transmission electron microscope (TEM) that were associated with both ruptured adipocytes and reactive macrophages within the thoracic aorta PVAT. The presence of approximately 60–70 nanometer small extracellular vesicle adipose-derived and macrophage-derived exosomes were also identified (Figure 4) [32,33]. 

These small EVexos are not only capable of regional paracrine signaling to increase PVAT inflammation but also are capable of long-distance endocrine signaling to signal the NVU BBB BECs. Furthermore, these endocrine signaling EVexos may contain proinflammatory cytokines and chemokines in addition to proinflammatory micro RNAs, which result in NVU BBB BEC signaling to induce neuroinflammation as previously presented and discussed in Figure 1 [32,33]. 

In the following Section 2 and Section 3, a possible unifying hypothesis will be discussed wherein the NVU BBB BECs that are activated by sLPS and lpsEVexos, and ***p***CC from the gut microbiota dysbiosis regions and ***p***CC from the excessive VAT depots play a central role in the development of EPVS in the MetS. Dual signaling of the BECs via ***p***CC and proinflammatory sLPS and lpsEVexos result in NVU BBB BEC activation and dysfunction (BECact/dys) as presented in Figure 1 with increased permeability via paracellular and transcellular mechanisms, which contribute to the development of EPVS. These lpsEVexos and ***p***CC mechanisms will act in synergy along with obstruction of the PVS due to inflammation, vascular stiffening with impaired arterial-arteriole pulsatility, neuronal or myelin atrophy, and capillary rarefaction. However, in this review, the main focus is on BBB disruption via LPS with increased transcytosis and fluid volume increase in the post-capillary PVS that contributes to enlarged perivascular spaces (EPVS). Importantly, BEC signaling of TLR4 also contributes to the increase in PVS inflammation that, in turn, results in the obstructive outflow from the PVS conduit to the subarachnoid space (SAS) cerebrospinal fluid (CSF). In addition, BBB disruption and NVU uncoupling could also contribute to regional brain atrophy and decreased microvascular pulsatility. In the following sections, the term BEC*act/dys* will be used; therefore, it is appropriate to define this term in the context of this paper. BEC*act* will be defined by the increased BEC expression of cell-surface adhesion molecules, such as vascular cell adhesion molecule 1(VCAM-1), intercellular adhesion molecule 1 (ICAM-1), and endothelial leukocyte adhesion molecule (ELAM, also known as E-selectin) creating a proinflammatory BEC. BEC*dys* will be defined as the decreased synthesis, release, and/or activity of endothelium-derived nitric oxide (NO) [33,34]. 

## 2. sLPS and lpsEVexos Activate BECs via TLR4

Both adipose and macrophage-derived EVexos of the VAT are important players in regard to paracrine and endocrine long-distance signaling and are known to participate in metabolic and immune signaling [35]. LPS is a glycolipid (Lipid A) polysaccharide and lipids; glycolipids may be included in adipose-derived EVexos along with other lipids, proteins, and nucleic acids, including multiple microRNAs (miRNAs), long non-coding RNAs (lncRNAs), messenger RNAs, circular RNAs. However, the lipid content adiEVexos have been understudied, and current lipidomic studies are scarce [35]. Currently, it is not known if LPS from adipocyte rupture and death or reactive MΦ are capable of generating or secreting lpsEVexos.

Peripherally-derived LPS from gut dysbiotic microbiota are incorporated into small lpsEVexos peripherally, and these lpsEVexos are then transported via the systemic circulation to activate target BEC TLR4 (Figure 2 and Figure 5) [17,33].

Similarly, it is now known that small EVexos contain multiple microRNAs (miRNAs) in obesity, MetS, and T2DM that are derived from the VAT depot adipocytes and CLS macrophages that cause metainflammation in the periphery [33]. Importantly, the miRNA profiles (multiple associated miRNAs) generated might be even more important than just the effects of a single miRNA from these depots (Figure 6) [33].

In the near future, we will begin to understand the role of the microRNAs (miRNAs) carried by the small EVexos that are known to arise from the dysbiotic gut microbiome, similar to how we have learned about the small EVexos miRNAs that are secreted from the VAT adipocytes and macrophages. It will be interesting and advance the science in this field as we learn more about how the miRNAs profiles carried by small EVexos from the microbiome dysbiotic gut and the VAT depots and how they may affect the BBB BECs to signal the BEC TLR4, in addition to they may interact with the ***p***CC uptake by BECs to result in BEC*act/dys* and contribute to the development of EPVS [17,33]. 

Once TLR4 is signaled by sLPS and/or lpsEVexos, there is a cascade of signaling mechanisms, which involve phosphatidylinositol 3-kinase/protein kinase B (PI3/AKT) and mitogen-activated protein kinase (MAPK) signaling to activate nuclear factor kappa B (NF-κB) that results in increased proinflammatory cytokines tumor necrosis factor alpha (TNF-α), interleukin-6 (IL-6), and the chemokine C-C Motif Chemokine Ligand 5 (CCL5) also referred to as regulated on activation normal T expressed and secreted (RANTES). CCl5 (RANTES) results in the attraction of microglia cell(s) (MGCs) to the NVU BEC*s* and is associated with increased macro- and micro-pinocytotic vesicles resulting in increased permeability [5] as occurred in the same CD-1 male mice at 20 weeks of age utilizing the same LPS protocol as in a previous study [6] (Figure 7).

Increased NVU BEC*act/dys* (associated with statistically increased pinocytotic vesicles) is known to increase permeability, which is thought to be due to LPS-induced disruption of the NVU blood-brain barrier (BBB) with increased transcellular and paracellular leakage. For example, Banks et al. measured (14)C-sucrose, radioactive albumin, and TEER, which demonstrated increased permeability and pointed strongly to the dysfunction of both transcytotic and paracellular pathways to result in increased permeability [6]. Further, this disruption of the NVU BBB would allow increased fluid flow into the post-capillary PVS that may promote EPVS [5,6].

There were multiple TEM remodeling changes found to be present in BECs when activated by LPS administration, which included increased BEC transcytotic vesicles consisting of both macro- and micro-pinocytosis as in Figure 5, plasma membrane ruffling, increased EVexosomes, and extracellular microvesicles-particles [36,37]. Additional characteristics of BEC*act/dys* included increased leukocyte, platelet, and RBC adhesion [30]; increased upregulation of endothelial intercellular adhesion molecule 1/vascular cell adhesion molecule 1 (ICAM-1/VCAM-1) receptors; increased BEC stiffening by atomic force microscopy; EC contraction with shortening, lifting, and separation [38,39,40]. 

### Metainflammation, Peripheral Cytokines and Chemokines (pCC) Derived from Gut Microbiota Dysbiosis, and VAT Crown-Like Structures (CLS) Concurrently Result in BECact/dys and Neuroinflammation

Additionally, obesity (especially VAT) that associates with the MetS is now considered to represent a state of chronic low-grade, sterile inflammation, which is commonly referred to as metainflammation [14,41,42]. The metainflammation that occurs in both the VAT adipose depots and in the gut dysbiotic microbiota regions is capable of generating increased ***p***CC that are transported to the NVU BECs. These ***p***CCs are capable of signaling the BECs concurrently with sLPS and lpsEVexos to act synergistically and result in neuroinflammation and EPVS (Figure 8) [1,33]. 

Notably, aging, immunosenescence, inflamm-aging (inflammatory aging), and metainflammation make a significant contribution not only to the aging process but also to the development of neuroinflammation and the known increase in EPVS in aging and its association with age-related neurodegenerative diseases [43].

Chronic peripheral inflammation (metainflammation)—hyperinflammation, as occurs in the MetS (Figure 3)—plays a very important role in BEC*act/dys* (Figure 2). Indeed, dual peripheral signaling by VAT-derived***p***CC and gut microbiota dysbiotic pCC, sLPS, and lpsEVexos are crucially important in the development of neuroinflammation and increased ***CNS***CC as a result of BECa*ct/dys*. In turn, this neuroinflammation (***CNS***CC) plays a huge role in the activation of the brain injury and response to injury wound healing process (hemostasis, inflammation, proliferation, and remodeling phases) which results in capillary rarefaction as well as the proliferation of perivascular macrophages (MΦ) within and immediately adjacent to the perivascular space. This accumulation of perivascular macrophages and other inflammatory cells and their accumulated phagocytic debris causes a stalling, stagnation, and obstruction of PVS fluid flow, which results in proximal obstructive dilation and the development of EPVS as determined by MRI and recent TEM studies discussed in the following Section 3 (Figure 9, Figure 10 and Figure 11). Thus, the author feels increasingly confident that the inflammatory phase of the wound healing process may be the most influential phase of the response to injury wound healing mechanism remodeling in MetS, obesity, and T2DM. 

## 3. From BEC*act/dys* to EPVS to SVD, to Accelerated Brain Aging, to Increased Vulnerability, to Impaired Cognition, to Neurodegeneration, to Dementia and Age-Related Neurodegenerative Diseases

NVU BEC*act/dys* occurs primarily as a result of ***p***CC-derived metainflammation signaling, and we now know that this includes sLPS and lpsEVexos from the gut microbiota dysbiosis in obesity and MetS [17]. BEC*act/dys* is known to be a key player in the development of EPVS due to dual signaling events via metainflammation from gut dysbiosis regions and VAT depots. EPVS are thought to be caused primarily by the following events: *i*. Obstruction due to chronic perivascular inflammation that is instigated by BEC*act/dys* with BBB disruption resulting in not only obstruction but also stagnation of flow due to excessive amounts of waste material to decrease flow within the PVS; *ii*. Increased proteins and fluid coming into PVS due to increased permeability of BBB due to BBB disruption of tight and adherens junctions (TJ/AJ) with increased paracellular influx or by the transcytotic route via increased transcytosis of both micro- and micro-pinocytotic vesicles of the activated BECs (Figure 5); *iii*. Decreased fluid outflow from PVS due to impaired or dysfunctional astrocyte end-feet due to detachment or separation from the NVU with decreased fluid uptake by the aquaporin-4 (AQP4) water channels allowing fluid to accumulate in the PVS; *iv*. Vascular stiffening (arteriolar or venular) with decreased pulsatility and deceased flow that is associated with not only aging but also underlying extra and intracranial atherosclerosis along with arteriolosclerosis; *v*. Myelin loss and neuronal atrophy in addition to those yet to be identified and fully studied, such as the role of capillary rarefaction in allowing for increased PVS fluid volume due to a chronic CNS wound healing response to injury by *p*CC, sLPS, and lpsEVexos with the development of EPVS as found in obesity, MetS, T2DM, and age-related neurodegenerative diseases (Figure 8) [1,21,24,25,33,44,45,46,47,48,49,50,51,52,53]. EPVSs are known to be associated with and are a biomarker for SVD. In Turn, SVD and EPVS are associated with accelerated brain aging due to repeated brain injury and the response to injury wound healing (Figure 9). 

These repeated injuries and response to injury wound healing mechanisms result in increased remodeling, including EPVS, and are associated with increased vulnerability to further injury, and a vicious cycle is triggered. 

EPVSs are known to be a biomarker and associate with SVD once they are large enough to be identified by T2-weighted MRI, and they are also known to associate with WMH and Lacunes [54]. PVSs on MRI have a signal intensity similar to CSF, which suggests the fluid within PVSs and CSF are comparable. The identification of PVS on MRI actually depends on the ISF and CSF within the PVS rather than the surrounding cellular tissues. The fluid within PVS may contain a proteinaceous material, such as fibrin/fibrinogen and ECM materials, that allow it to have signal intensities similar to the CSF [54]. Considering the injury to the BEC by ***p***CC, peripheral sLPS, lpsEVexos, and the response to injury wound healing mechanisms in the CNS (Figure 7), it is not difficult to consider the development of capillary rarefaction as related to EPVS (Figure 10) [1]. 

Recently, the author was able to identify a TEM image of an EPVS in an LPS intraperitoneal-treated male CD-1 mouse at 20 weeks of age [5]. This single image not only depicts an enlarged perivascular space but also allows one to view a resident reactive macrophage (rMΦ) within the EPVS. This rMΦ contributes to perivascular and, more specifically, intra-PVS inflammation, which would result in the stagnation of interstitial fluid flow and obstruction, which is one of the main contributors to the development of EPVS (Figure 11) [1,55]. 

In health, true capillaries without a PVS are responsible for the uptake of nutrients, water, oxygen, and metabolic efflux, while the post-capillary venule with its PVS serves as a conduit for interstitial metabolic waste disposal via the PVS glymphatic system. In addition, this PVS allows for the transmigration of immune cells (innate and adaptive leukocytes) into the CNS parenchyma, as described by Owens et al. [55]. This strongly implicates the PVS in the two-step process of neuroinflammation, with step one consisting of circulating immune cells rolling, adhering, and transmigration across the BECs of post-capillaries venules and, step two, the progression of immune cells across the PVS and migration across the ACef or glia limitans into the CNS parenchymal interstitium [55]. After leukocytes cross the BEC BMs, they enter the post-capillary venule PVS, where they are thought to be assisted by the resident perivascular resident macrophage or dendritic cells within the PVS by their secreted MMPs (MMP-2 and MMP-9) to allow for the penetration of the ACef BM (glia limitans) to migrate into the CNS parenchyma and thus complete the second step of neuroinflammation [55]. 

Additionally, post-capillary venular PVSs are responsible for waste removal by what is now referred to as the glymphatic system [56], which is responsible for the drainage of metabolic waste from the ISF to the SAS to the CSF and venous drainage. If obstructed, the resulting EPVS will result in its identification by MRI and will allow even greater amounts of fluid and neurotoxic proteins to accumulate with ongoing perivascular neuroinflammation and decreased waste efflux [56]. 

Just as EPVSs are known to be biomarkers for SVD, numerous biomarkers have also been identified for BEC*act/dys* and SVD. Circulating biologic markers of BEC*act/dys* have been identified and currently can be measured in the laboratory [57]. These elevated biomarkers include some of the following: adhesion molecules (I-CAM, V-CAM, and selectins); asymmetric dimethyl arginine; C-reactive protein and highly sensitive CRP; D-dimer; endothelial progenitor cell(s); fibrinogen; hyperhomocysteinemia; lipoprotein little a; lipoprotein phospholipase-A2; malondialdehyde; MMPs; myeloperoxidase; plasminogen; plasminogen activator inhibitor type 1; plasmin-alpha2-plasmin inhibitor complex; ***p***CC (numerous); thrombin-antithrombin complex; tissue factor; tissue factor pathway inhibitor; thrombomodulin; vascular endothelial growth factor; tissue plasminogen activator; Von Willebrand factor [57]. 

The complicated possible sequence of events that lead to the formation of enlarged perivascular spaces (EPVS) and the evolutionary spectrum over time to progress from EPVS to SVD, neuroinflammation, impaired cognition, and neurodegeneration still have a great number of gaps to be filled with continuing research; however, the following may serve as an early pathway for discovery (Figure 12) [1]. 

While this review has focused primarily on the post-capillary venule glymphatic efflux waste clearance system, it is important to note that the arterial influx system via pia arteries and pre-capillary arterioles to deliver CSF to the interstitium is essential. Importantly, this arterial influx system not only delivers CSF to the interstitium of the brain but also plays a role in waste clearance via the intramural periarterial drainage (IPAD) efflux waste clearance system, especially in cerebral amyloid angiopathy [58].

### Interaction of BECact/dys and Aquaporin-4 (AQP4) in EPVS

One would be remiss if the role of aquaporin 4 (AQP4) was not included in the discussion of aberrant glymphatic function and EPVS because of the intimate contact and crosstalk between the NVU BEC/Pc shared BM and the plasma membrane of the neuroglial ACef (Figure 11A). AQP4 water channels are small hydrophobic proteins (~30 kDa monomer) that facilitate bi-directional water transport in response to osmotic gradients to maintain fluid homeostasis between the post-capillary PVS and the lining ACef [58,59,60,61,62]. Additionally, they are also known to be involved in diverse functions, such as regulation of extracellular space volume, potassium buffering, cerebrospinal fluid circulation, interstitial fluid resorption, waste clearance, neuroinflammation, osmoregulation, cell migration, and Ca^2+^ signaling. [59,60,61,62,63] (Figure 13). 

In addition to the role of BEC*act/dys,* AQP4 is also involved with the development of EPVS [60,61,62,63]. BEC*act/dys* leads to activation (inflammation), dysfunction (decreased NO bioavailability), and blood-brain barrier (BBB) disruption, which increases NVU BBB permeability and neuroinflammation with increased resident, reactive intra-PVS macrophages (rMΦ) (Figure 11B) [1,5,55]. Concurrently, AQP4 relocation from astrocyte end-feet (ACef) plasma membranes (due to loss of polarization) contributes to EPVS [59,61,62,63]. Additionally, ACef detachment and retraction are known to occur in obese, insulin-resistant (IR) diabetic *db/db* models, which could interfere with the normal function of polarized AQP4 capable of homeostatic fluid control [63]. In addition, it is known that dysfunction of AQP4 exists, such as occurring due to the formation of autoantibodies to AQP4 in neuromyelitis optica spectrum disorder (NMOSD) with EPVS [60]. Indeed, the aberrant function of AQP4 water channels will contribute to the development of EPVS since AQP4 water channels facilitate the bidirectional flow of water and play a central role in brain fluid homeostasis, acting as a water channel protein. Both glymphatic influx of CSF and interstitial solute and waste clearance are dependent upon intact and functional perivascular AQP4 water channels [61,62]. Therefore, if AQP4 is relocated from ACef plasma membranes or if ACef is detached and retracted, this would interfere with water uptake from the increased water in PVS due to increased permeability and thus allow water to accumulate and result in the development of PVS fluid expansion and EPVS development. Further, recent MRI imaging of glymphatic function suggests that glymphatic function in humans is impaired in the presence of small vessel disease [26,60,61,62,63].

## 4. Conclusions

BECs are the first CNS cells to come into contact with peripherally derived neurotoxins capable of inducing BEC*act/dys* and neuroinflammation. Notably, BECs may be considered the sentinel cell of the brain’s innate immune system and are capable of antigen presentation. In obesity, MetS, and T2DM, the BECs are chronically signaled by peripherally-derived proinflammatory signals due to metainflammation that result in BEC*act/dys*. This review has focused primarily on sLPS and lpsEVexos, and ***p***CC from the regional proinflammatory gut microbiota dysbiosis and VAT depots [Figure 3, Figure 7, Figure 8, and Figure 10B]. Furthermore, these two peripherally-derived proinflammatory mechanisms (***p***CC and sLPS and lpsEVexos) with separate signaling mechanisms are each capable of instigating BEC*act/dys* that may even act synergistically due to their dual signaling effects on BECs to result in BEC*act/dys*. In addition, the dual signaling (***p***CC and LPS from the VAT and regional gut microbiota dysbiosis) sets in motion a three-step mechanism to result in EPVS as follows: Step (1) BEC*act/dys*; Step (2) neuroinflammation with excessive ***CNS***CC; Step (3) excessive ***CNS***CC result in reactive resident PVS macrophages producing excessive debris and excessive phagocytosis resulting in obstruction and dilation of PVS to promote the development of EPVS. In addition, Steps 1–3 may not only result in a self-perpetuating vicious cycle but also, along with the BECs’ increased permeability (due to BEC*act/dys*), may combine to result in EPVS. 

EPVSs are known to be biomarkers for SVD, and since the prevalence of SVD increases with age, there is now a consensus of opinion that the prevalence of EPVS-associated dementia will increase as life expectancy lengthens. This observation places EPVS and SVD as very important to the well-being of our younger society members, with the multiple risk factors and variables associated with the MetS as they age. Additionally, this also places our older society members globally at risk for accelerated brain aging, increased vulnerability to progressive neurodegeneration, impaired cognition, neurodegeneration, dementia, and age-related neurodegenerative diseases.

This review has examined the role of (1) peripheral inflammation/metainflammation and how it instigates CNS neuroinflammation; (2) sLPS and lpsEVexos while examining the important role they play in the development of BEC*act/dys* and EPVS; (3) how sLPS and lpsEVexos specifically signal the BBB BECs via the TLR4 and its downstream signaling pathway to result in neuroinflammation and BEC*act/dys* [Figure 2]; (4) both the VAT with its pCC and EVexos with their miRNAs and gut microbiota dysbiosis with its sLPS and lpsEVexos; (5) the MetS incorporating obesity (hyperlipidemia), hyperinsulinemia (IR/LR), HTN, and hyperglycemia (IGT, prediabetes, and +/−T2DM [Figure 3]; (6) VAT/PVAT and EVexos with TEM findings [Figure 4]; (7) VAT/PVAT EVexos and gut microbiota dysbiosis EVexo synthesis with the incorporation of miRNAs [Figure 5 and Figure 6]; (8) BEC*act/dys* with increased transcytosis and BBB permeability due to increased macro- and micro-pinocytic vesicles [Figure 7]; (9) gut microbiota dysbiosis and triangulation of metabolic, gut dysbiosis, and MetS [Figure 8]; (10) EPVS causation; (11) brain injury and the response to injury wound healing mechanisms [Figure 9]; (12) capillary rarefaction [Figure 10]; post-capillary venules and resident macrophages EPVS [Figure 12]. Clinically, EPVSs are an emerging functional and structural remodeling change that can be identified by non-invasive magnetic resonance imaging and are a known biomarker of SVD. EPVS are also being increasingly viewed as a marker of cerebrovascular pathology and neurodegeneration. Further, SVD is a leading cause of stroke and dementia globally, with their associated burdens to families and societies. 

In summary, this review has placed together many pieces of the puzzle as to how BECs play a central and key role in the development of EPVS in obesity, MetS, and T2DM.

## Figures and Tables

**Figure 1 medicina-59-01124-f001:**
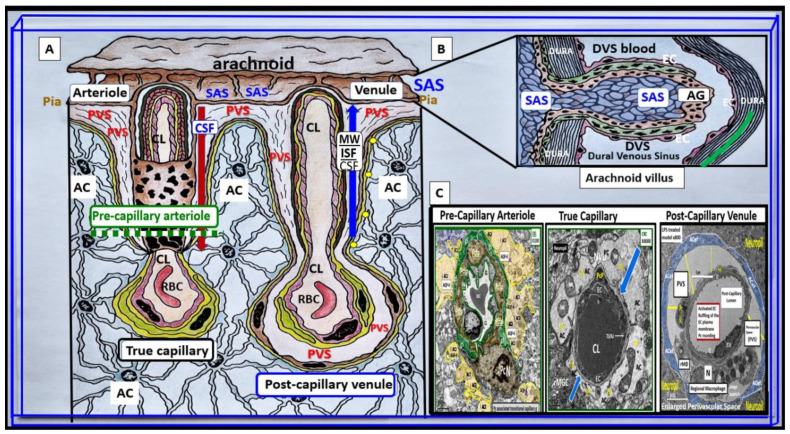
Illustration to provide a better understanding of the origins of the perivascular spaces (PVS) and their relationship to precapillary arterioles, true capillaries, and post-capillary venules. (**A**) demonstrates the relationship of the perivascular spaces (PVS) to the precapillary arterioles with the vascular lumen influx of oxygen, nutrients, solutes (red arrow) and the influx of CSF via the PVS (red arrow) to post-arteriole transitional zone capillaries (green dashed line), true capillaries with the vascular lumen delivery of oxygen, nutrients, and solutes, and the post-capillary venules that remove metabolic waste via the vascular lumen (carbon dioxide plus others) and PVS that is responsible for the removal of the admixture of ISF and CSF, and metabolic waste (MW including neurotoxins of misfolded proteins such as toxic amyloid beta and tau neurofibrillary fibrils) (blue arrow) to the SAS and arachnoid villus to the dural venous sinus (DVS). (**B**) demonstrates how the contents of the PVS drain into the subarachnoid space (SAS) and then into the dural venous sinus (DVS) via the arachnoid villi. (**C**) demonstrates the corresponding cross-sectional transmission electron micrographs of corresponding labeled capillary images in panel A. Importantly, note that the pia membrane of the glia limitans abruptly ends at the true capillary and that the astrocyte end-feet directly abuts the BEC basement membrane (blue arrows). This adapted and modified image is provided with permission by CC 4.0 [1]. AC = astrocyte; ACfp = astrocyte foot process-end feet; AQP4 = aquaporin four; Cl = capillary lumen; CSF = cerebrospinal fluid; EC = brain endothelial cell; ISF = interstitial fluid; N = nucleus; Pc = pericyte; PVS = perivascular space; Mt = mitochondria; RBC = red blood cell; rMΦ = resident reactive macrophage.

**Figure 2 medicina-59-01124-f002:**
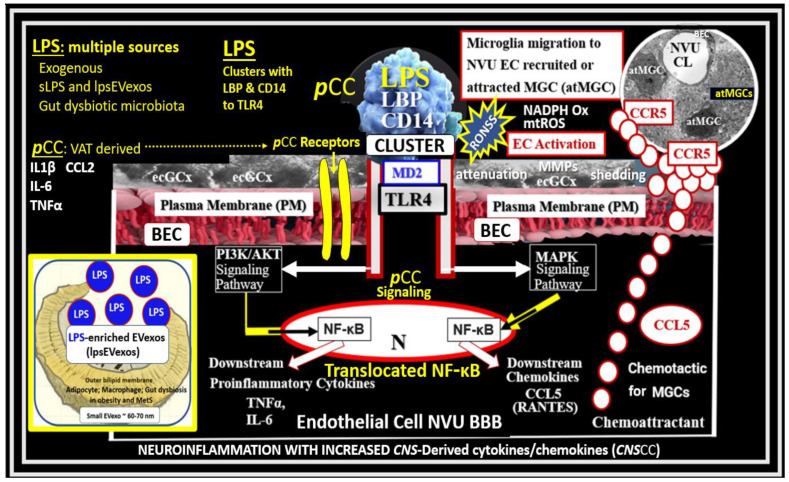
Possible dual signaling mechanisms illustrating how peripherally-derived soluble lipopolysaccharide (sLPS), small extracellular LPS vesicle exosomes (lpsEVexos), and peripherally-derived cytokines and chemokines (***p***CCs) result in brain endothelial cell(s) (BEC) activation and dysfunction (BEC*act/dys*) to result in neuroinflammation, blood-brain barrier (BBB) disruption, and enlarged perivascular spaces (EPVS) in obesity, insulin resistance (IR), and the metabolic syndrome (MetS). sLPS and lpsEVexos are known to induce increased immunomodulation of target BECs. There are multiple peripheral sources of LPS, which are capable of signaling and activating BECs. Peripheral lpsEVexos are derived primarily from gut dysbiotic microbiota and leaky gut syndrome, while ***p***CCs are derived from excessive visceral adipose tissue accumulation from VAT crown-like structure macrophages. Importantly, sLPS and lpsEVexos plasma levels are known to be elevated in obesity and MetS. sLPS and lpsEVexos signal toll-like receptor 4 (TLR-4) via a cluster of LPS, lipoprotein binding protein (LBP), and a cluster of differentiation 14 (CD14), which along with myeloid differentiation factor-2 (MD2) initiates TLR4 proinflammatory signaling. Importantly, note that increased ***p***CCs can similarly signal BECs to produce BEC-derived central nervous system cytokines/chemokines (***CNS***CC). TLR4 signaling sets in motion a cascade of signaling events that ultimately attracts microglia cell(s) (MGCs) to the NVU to result in neuroinflammation in addition to BEC increased macro-and micro-pinocytotic vesicles that could result in increased NVU BEC permeability. This figure also demonstrates that activated BECs (aBECs) are capable of secreting the chemokine CCL5/RANTES to attract microglia cells (atMGCs) to the NVU BEC via activation of BEC toll-like receptor 4 (TLR4). Additionally, note that the signaling of TLR4 is responsible for the signaling of nuclear factor kappa B (NF-κB) via the PI3K/AKT and MAPK signaling pathways. Additionally, NF-κB also induces BEC fractalkine (CX3CL1), which is capable of increasing proinflammatory changes in aBEC, which result in neuroinflammation. Importantly, note that the endothelial glycocalyx (ecGCx) may be partially shed or thinned, which associates with BEC*act/dys*. BEC dysfunction results in decreased nitric oxide (NO) bioavailability that contributes to NVU uncoupling and even further shedding of the ecGCx. Importantly, signaling of TLR4 by LPS also causes increased BEC reactive oxygen, nitrogen, and sulfur species (RONSS) via nicotinamide adenine dinucleotide phosphate oxidase reduced (NADPH oxidase) that are capable of activating matrix metalloproteinases (MMPs), which further results in ecGCx shedding as well as tight and adherens junction impairment. atMGC = attracted microglia cell; CCL5 = chemokine (C-C motif) ligand 5 (chemoattractant for microglia cells); CD14 = cluster of differentiation 14; CCR5 = C-C chemokine receptor type 5, also known as CD195; CL = capillary lumen; EC = endothelial cell, brain endothelial cells; ecGCx = endothelial glycocalyx; EV = extracellular vesicles; EVMp = EVmicroparticles or microvesicles; EVexo = EVexosomes; IL1-β = interleukin-1 beta; IL-6 = interleukin-6; LPS = lipopolysaccharide; LBP = lipopolysaccharide-binding protein; MAPK = mitogen-activated protein kinase; Mp = microparticles; N = nucleus; PI3K/AKT = phosphatidylinositol 3-kinase / protein kinase B; PM = plasma membrane; RANTES = regulated on activation, normal T cell expressed and secreted; TNFα = tumor necrosis alpha.

**Figure 3 medicina-59-01124-f003:**
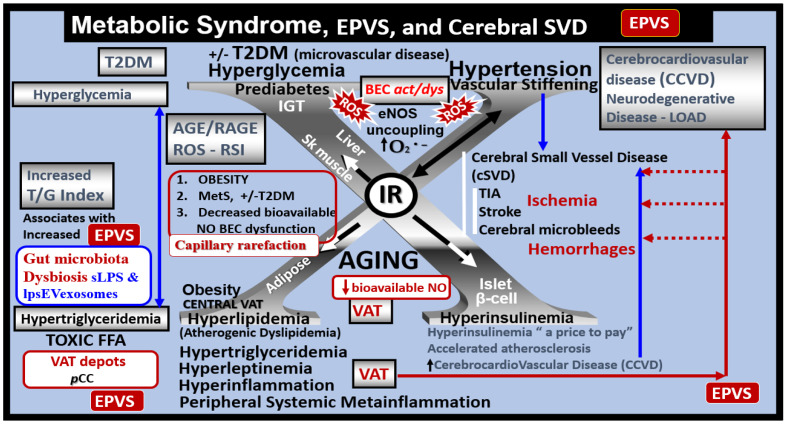
Metabolic syndrome (MetS), enlarged perivascular spaces (EPVS), and cerebral small vessel disease (cSVD). Note how the accumulation of visceral adipose tissue (VAT), obesity, and hyperlipidemia (atherogenic dyslipidemia) appear to drive the MetS and the other three arms of the letter X, which includes insulin resistance (IR) and the associated hyperinsulinemia, hypertension; vascular stiffening, and hyperglycemia with or without manifest type 2 diabetes mellitus (T2DM). Note the prominent closed red arrows emanating from VAT to cerebrocardiovascular disease (CCVD), SVD, TIA, Stroke, and cerebral microbleeds and hemorrhages. Ensuing brain endothelial cell(s) (BEC) activation and dysfunction (BEC*act/dys*) with its proinflammatory and prooxidative properties result in endothelial nitric oxide synthesis (eNOS) uncoupling with increased superoxide (*O_2_ ^• –^*) and decreased nitric oxide (NO) bioavailability. Importantly, note that obesity, MetS, and decreased bioavailable NO interact to result in capillary rarefaction that allows the PVS to undergo expansion and dilation to become enlarged perivascular spaces (EPVS), which are biomarkers for cerebral small vessel disease (SVD). AGE = advanced glycation end-products; RAGE = receptor for AGE; AGE/RAGE = advanced glycation end-products and its receptor interaction; βcell = pancreatic islet insulin-producing beta cell; FFA = free fatty acids—unsaturated long chain fatty acids; IGT = impaired glucose tolerance; LOAD = late-onset Alzheimer’s disease; ROS = reactive oxygen species; RSI = reactive species interactome; Sk = skeletal: TG Index = triglyceride/glucose index; TIA = transient ischemia attack.

**Figure 4 medicina-59-01124-f004:**
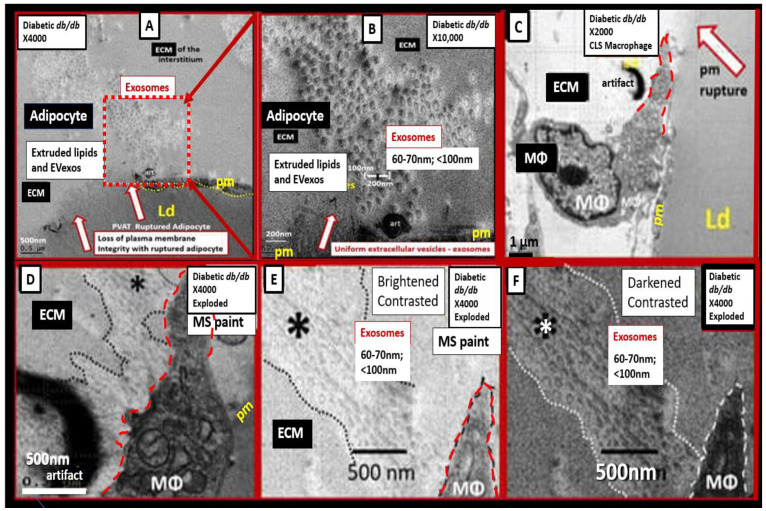
Perivascular adipose tissue (PVAT), hypertrophic adipocyte, and macrophage-derived small extracellular vesicle exosomes from the obese, insulin-resistant, and diabetic female *db/db* mouse models at 20 weeks of age. (**A**) depicts a ruptured plasma membrane (pm) (open red arrows) of a hypertrophic differentiated unilocular white adipose tissue (WAT) adipocyte as compared to control models with brown adipose tissue (BAT) (not shown). Note the extruded lipid contents via the adipocyte’s ruptured pm (dashed boxed) and surrounding extracellular matrix (ECM). (**B**) is a higher magnification of the boxed-in region of panel A that allows for greater clarity of the presence of 60–70 nm uniform sphere-shaped vesicles that are representative of small extracellular exosomes (EVexos). (**C**) depicts a representative crown-like structure (CLS) adherent macrophage (MΦ) to the pm of a ruptured adipocyte in the PVAT from the same model in the same image collection data set. Note that the upper tip of the MΦ is outlined in red-dashed lines to illustrate the regions where EVexos will be identified in greater clarity in panels D, E, and F. (**D**) depicts a higher magnification and exploded image of the MΦ depicted in panel C. Note the region demarcated by the black dashed line and asterisk that depicts possible EVexos, which are highlighted in panels E and F. (**E**) depicts and even greater exploded image of panel D that is brightened and contrasted in Microsoft (MS) Paint in order to further demonstrate the uniform 60–70 nm spheres representative of MΦ-derived EVexos (asterisk). (**F**) depicts a darkened contrasted image of panel E in order to better demonstrate the presence of EVexo within the CLS MΦ. Permission to reproduce these modified images is provided by CC 4.0 [32,33]. Magnification and scale bars appear in each panel. Ld = lipid droplet.

**Figure 5 medicina-59-01124-f005:**
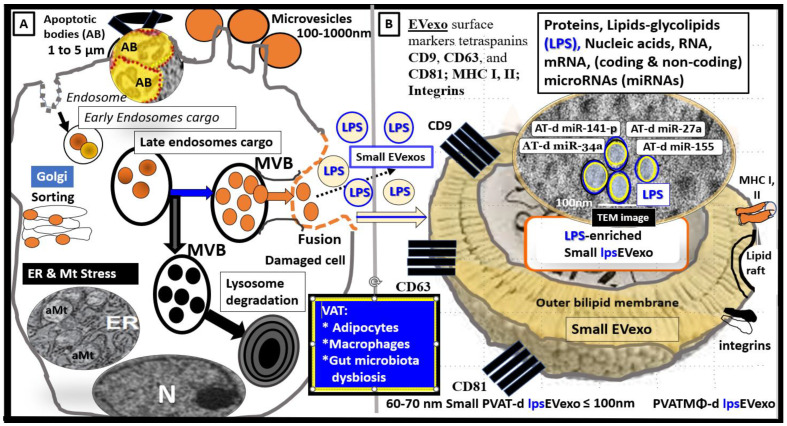
Illustration of peripheral cells and gut microbiota that are capable of generating lipopolysaccharide (LPS) enriched small extracellular vesicle exosomes (lpsEVexo). (**A**) illustrates the formation of multivesicular bodies and subsequent lpsEVexos. (**B**) illustrates a single small extracellular vesicle exosome. Permission to publish this modified image is provided by CC 4.0 [33]. CD = cluster of differentiation; d = derived; ER = endoplasmic reticulum; MHC = major histocompatibility complex; m = messenger; MVB = multivesicular bodies; N = nucleus; PVAT = perivascular adipose tissue; VAT = visceral adipose tissue.

**Figure 6 medicina-59-01124-f006:**
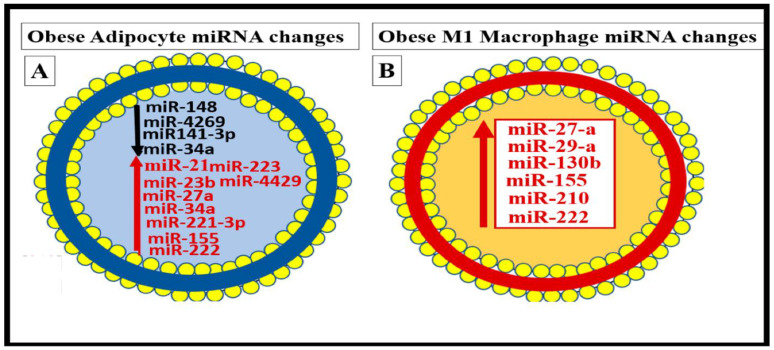
Cell-specific miRNA profiles in adipocytes and reactive, polarized M1-like macrophages (MΦs) in the obese, MetS, and T2DM visceral adipose tissue (VAT) depots. (**A**) depicts the decreased miRNAs with black lettering and a black arrow and increased miRNAs with red lettering and a red arrow in the obese VAT that includes perivascular adipose tissue (PVAT). (**B**) depicts the elevated miRNAs with red lettering and a red arrow in the reactive, polarized, crown-like structure M1-like MΦs in the VAT/PVAT. Image provided with permission by CC 4.0 [33].

**Figure 7 medicina-59-01124-f007:**
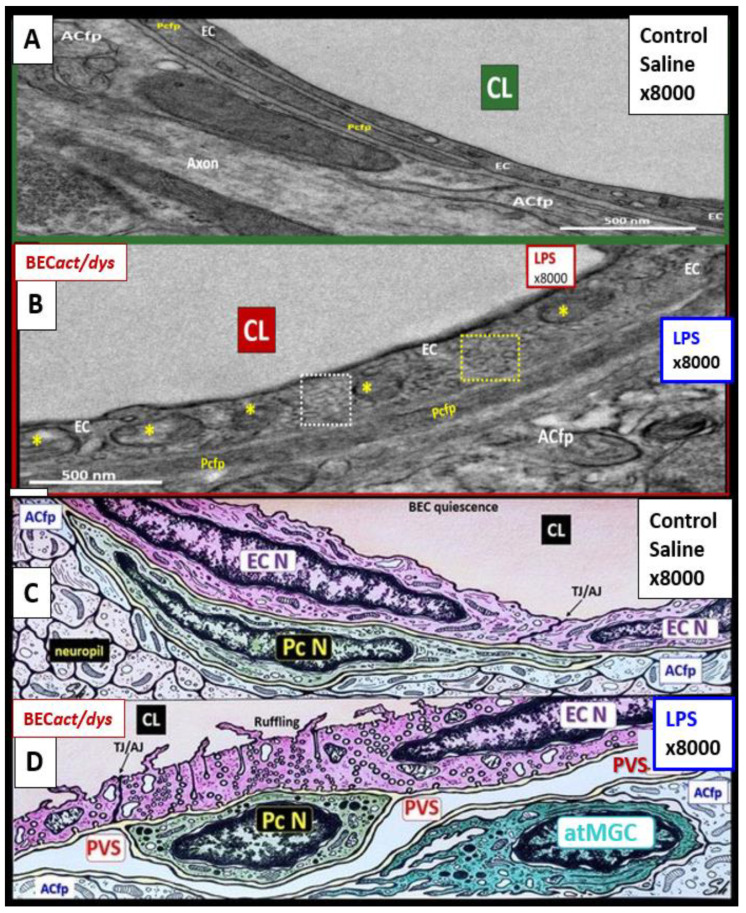
Transmission electron microscopy (TEM) and artistic rendition of increased pinocytosis in male CD-1 mice BECs treated with 3 mg/kg LPS. (**A**,**C**) demonstrate the normal ultrastructure of the BECs and note the paucity of pinocytotic vesicles. (**B**,**D**) depict the cellular remodeling in the BEC treated with LPS. Note the statistically significant increase in pinocytotic vesicles, both macro-pinocytic and micro-pinocytic vesicles. Additionally, these BECs depicted plasma membrane ruffling and pericyte rounding (not shown in this image) compatible with the activation of BEC and pericytes of the NVU. Through TEM studies, these mice did not depict any identifiable remodeling of the tight and adherens junctions (TJ/AJ). Importantly, this same male model at the same age of 20 weeks with the same LPS dose demonstrated increased permeability of the NVU and BBB. Magnification × 8000; scale bar = 500 nm in panels A and B, while panels C and D were not drawn to scale. Images provided with permission by CC 4.0 [5]. ACfp = astrocyte foot processes (end-feet); atMGC = attracted microglia cell; EC = endothelial cell/brain endothelial cell; EC N = endothelial/brain endothelial cell nucleus; Pcfp = pericyte foot processes; PcN = pericyte nucleus; PVS = perivascular space.

**Figure 8 medicina-59-01124-f008:**
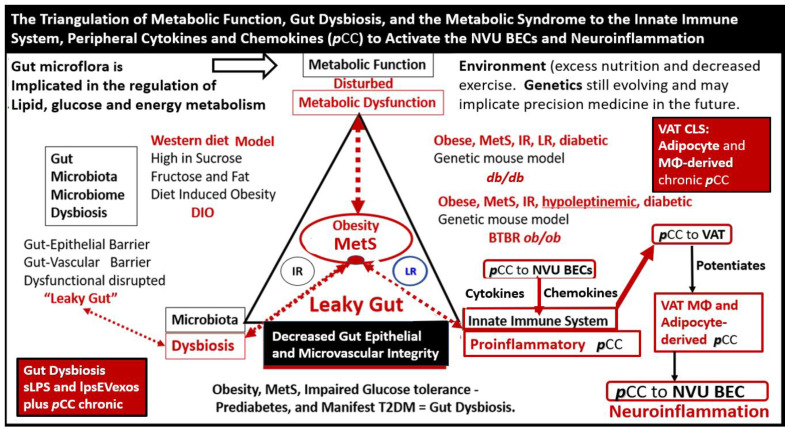
The triangulation of metabolic function, gut dysbiosis, and the metabolic syndrome to the innate immune system, peripheral, proinflammatory cytokines and chemokines (***p***CC) to signal the neurovascular unit (NVU) brain endothelial cell(s) (BECs) resulting in BEC activation and dysfunction (BEC*act*/*dys*) to result in neuroinflammation. This schematic demonstrates that gut microbiota dysbiosis, metabolic dysfunction, and an activated proinflammatory innate immune system are associated with obesity and metabolic syndrome (lower-left red box). Importantly, this triangulation also associates with the metainflammation that is produced in the obese visceral adipose tissue depots (upper-right red box). These two distinct sites of metainflammation (red boxes) and dual signaling of BECs will each signal the central nervous system brain endothelial cells to result in neuroinflammation to result in BEC*act/dys* and contribute to enlarged perivascular spaces EPVS. Importantly, this dual signaling by the gut and visceral adipose tissue (VAT) of BECs may be synergistic in obesity and the MetS. Db/db = obese, insulin resistance diabetic genetic mouse model; DIO = diet-induced obesity; BTBRob/ob = black and tan brachyuric ob/ob mouse model of obesity and diabetes; IR = insulin resistance; LR = leptin resistance; sLPS = soluble lipopolysaccharide; lpsEVexos = lipopolysaccharide extracellular vesicle exosomes; MΦ = macrophage; MetS = metabolic syndrome; ***p***CC = peripheral cytokines/chemokines; T2DM = type 2 diabetes mellitus.

**Figure 9 medicina-59-01124-f009:**
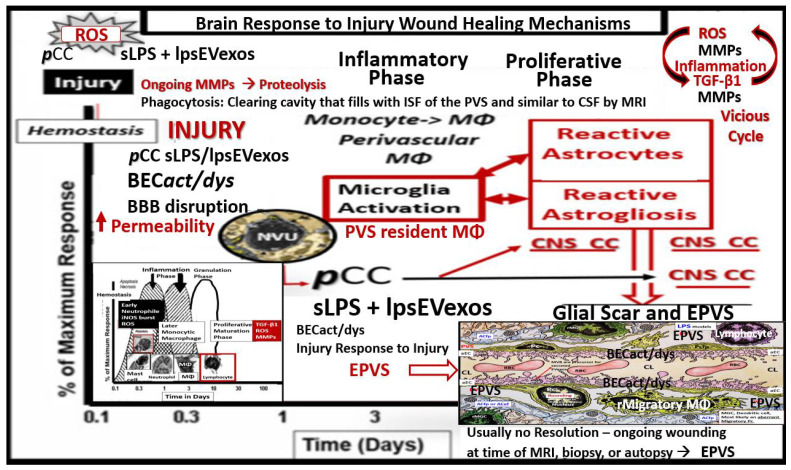
Response to injury wound healing mechanisms and enlarged perivascular spaces (EPVSs). In this case, the injury that triggers this response is the increased peripheral metainflammation with increased peripheral cytokines/chemokines and soluble lipopolysaccharide (LPS) and LPS extracellular vesicle exosomes (lpsEVexos), resulting in brain endothelial cell activation and dysfunction (BEC*act/dys*). This image depicts the response to injury wound healing mechanism in the central nervous system (CNS). Note the small insert (lower-left), which illustrates the timeline of response to injury wound healing mechanisms in peripheral tissues, such as in the skin or the lungs, and includes the various cells and mechanisms involved, which progress to fibrosis. Importantly, the injury, in this case, is the peripheral cytokines/chemokines (***p***CC) and the soluble LPS (sLPS) and LPS-enriched extracellular vesicle exosomes (lpsEVexos) that result in brain endothelial cell activation and dysfunction (BEC*act/dys*), which promotes ROS with further neuroinflammation and BEC*act/dys* with decreased NO that promotes even more superoxide-reactive oxygen species (ROS) and reactive oxygen, nitrogen species (RONS), reactive oxygen, nitrogen, sulfur species (RONSS), and the reactive species interactome (RSI). Importantly, this BEC*act/dys* will also promote an increase in BEC-derived TGF-β1 and possible TGF-β1 in reactive pericytes, reactive, and attracted microglia cells (rMGC) to instigate the vicious dual cycle of ROS → Inflammation via NFkappa B → TGF-β1 → ROS and the RSI → to activate local matrix metalloproteinases (MMPs) within the NVU and its PVS in its adjacent post-capillary venules. These combined mechanisms, such as stalling of waste removal flow and efflux due to increased accumulation of proteolytic debris that occurs in both the obese, MetS, and diabetic *db/db*, the female BTBR *ob/ob* and the CD-1 male mice treated with exogenous ip LPS. This ongoing production of ROS, Inflammation, and TGF-β1 will further aggravate the NVU BEC injury and, in turn, activate the genetically programmed response to injury wound healing mechanism (via embryonic genetic memory). Additionally, in the obese diabetic *db/db* and *ob/ob* models, the CNS injury may also be due to glucolipotoxicity and excessive RONSS of the RSI. The resident immune cell in the CNS is the neuroglial microglia cell (MGC). MGCs support the initial acute and chronic inflammatory responses to injury. However, the peripheral monocyte/monocyte-derived macrophages (MΦ) from the regionally activated NVUs are supportive not only to the resident MGCs but also to resident MΦs within the PVS to clear debris from CNS microthrombus/clot or microhemorrhage—stroke ischemic and hypoxic injury such that the peripheral inflammatory systems aid the resident MGC and the resident macrophages residing within the PVS that will also contribute to further injury and response to injury wound healing mechanisms. Notably, the NVUs-activated and reactive endothelial cells (ECs), pericytes (Pcs), and astrocytes (ACs) of the blood-brain barrier (BBB) may contribute to the CNS cytokines/chemokine (*CNS*CC) activation as well as in the choroid plexus, which forms the blood-cerebrospinal fluid barrier (BCSFB) and the meningeal arachnoid BCSFB. Importantly, note that the CNS tissues do not have fibrocytes/fibroblasts in the CNS parenchymal tissues and, therefore, do not form a classic fibrotic scar as in peripheral tissues such as the lung or skin in small lower left-hand side insert. Instead, the CNS utilizes the glial astrocytes (ACs) to undergo the process of astrogliosis in order to form a glial scar to protect the adjacent neuropil from the injury and damage caused by a micro thrombotic/micro hemorrhagic stroke. Unfortunately, the activated CNS glial cells add to the *CNS*CC (IL-6, IL-1β, TNFα, growth factor TGF-β1) (IL-10 anti-inflammatory)/chemokine (CXCL2 -MCP-1 + others) toxic inflammatory damage to the CNS. Importantly as the proteolysis of the cellular debris is being cleaned up and phagocytized by macrophages utilizing MMPs, plus this new space will be filled by the ISF of the PVS, allowing it to increase its volume and contribute to the formation of EPVS.

**Figure 10 medicina-59-01124-f010:**
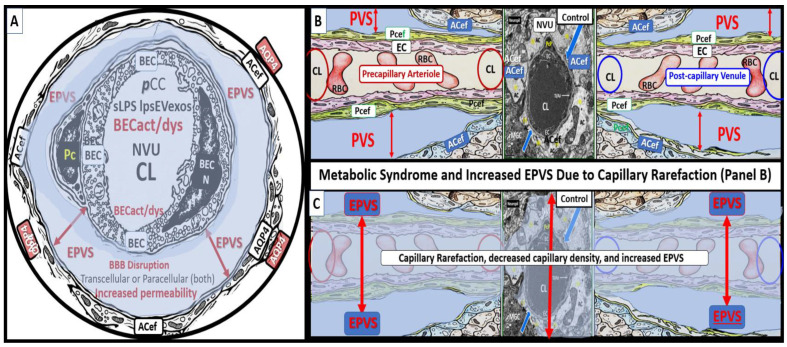
Capillary loss due to brain endothelial cell(s) BEC activation and dysfunction (BEC*act/dys*) with BBB disruption and neuroinflammation contribute to an increase in perivascular space(s) (PVS) fluid volume. (**A**) depicts a cross-section of a capillary with enlarged perivascular space(s) (EPVS) (double red arrows) as occurs in obesity, MetS, and exogenous LPS treatment in a post-capillary venule. Note the presence of the astrocyte end-feet (ACef) and the presence of aquaporin 4 (AQP4) and its translocation from the basal lamina of the ACef that will impair water uptake from the PVS. (**B**) demonstrates the normal appearance of a true capillary (middle), precapillary arteriole (left), and a post-capillary venule (right). (**C**) depicts capillary loss or capillary rarefaction as it occurs in obesity, MetS, and exogenous LPS treatment. Overall, this illustration depicts the digestion of capillaries within the perivascular spaces due to BEC*act/dys,* which occurs as a result of neuroinflammation. This neuroinflammation is triggered by peripheral cytokine and chemokine (***p***CC) and soluble lipopolysaccharide—lipopolysaccharide-enriched extracellular vesicle exosomes (LPS—sLPS-lpsEVexos) metainflammation and leaves a structural cavity-space void that is filled in by the surrounding PVS fluid and allows for an increase in total PVS fluid volume that could now be visualized by MRI. Furthermore, this illustration attempts to show this capillary loss by creating a semi-transparent masking overlay to depict the loss of the true capillary, precapillary arteriole, and post-capillary venules. Simplistically, EPVSs result from excess fluid “in” as a result of increased NVU permeability and decreased fluid and waste “out” (efflux) of the PVS due to obstruction to flow. BBB = blood-brain barrier; CL = capillary lumen; NVU = neurovascular unit.

**Figure 11 medicina-59-01124-f011:**
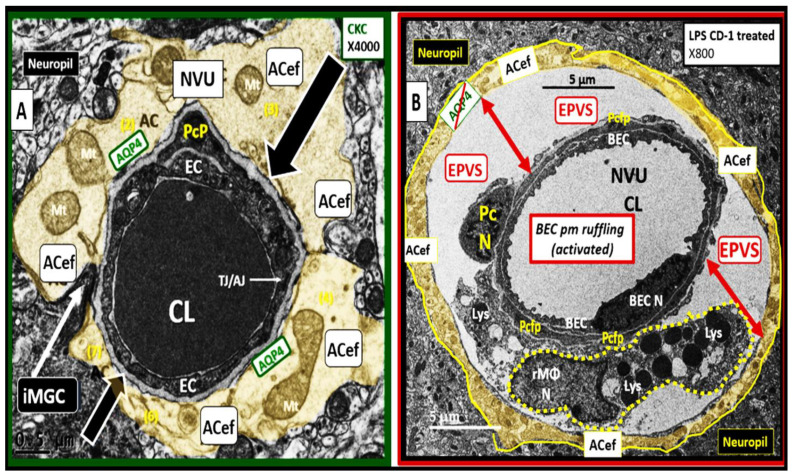
Lipopolysaccharide (LPS) treatment is associated with enlarged perivascular spaces (EPVS). (**A**) demonstrates a true capillary in a control non-LPS treated model. Note the tight abutment of the astrocyte end-feet (ACef) to the basement membrane (BM) of the neurovascular unit (NVU) brain endothelial cell (BEC)s and pericyte process end-feet or foot processes (PcP) (open black arrows). Additionally, note the aquaporin4 (AQP4) water channel that is located immediately adjacent to the NVU BM in the terminal ACef to assist in water uptake. (**B**) depicts a post-capillary venule that is known to have a space that is usually ≤ one micrometer, termed the perivascular space (PVS). Note the detachment and separation of the ACef from the NVU BM (double red arrows) along with the AQP4 water channel (crossed out to indicate dysfunction), which contributes to the widened PVS of up to 5 μm (double red arrows) to create EPVS. Importantly, note the large resident, reactive macrophage cell (rMΦ) (outlined with a yellow dashed line) with multiple lysosomes (Lys), vesicles, and vacuoles (suggesting polarization with increased reactivity) inferior to the NVU that is known to stain positive for CD-163 and is capable of antigen presentation to support ongoing perivascular and neuroinflammation that is also referred to as perivascular macrophage (CD-163 staining not shown). This intra-perivascular space macrophage is capable of migrating within the PVS and may contribute to EPVS as it digests debris within the PVS and contributes to stalling, stagnation, and obstruction of the interstitial fluid flow within the perivascular space, resulting in EPVS. Additionally, note the very close proximity of the rMΦ to the NVU pericyte foot process (Pcfp) and ACef, which allows for extensive crosstalk communication between these two cells. AC = astrocyte; CL = capillary unit; EC = brain endothelial cell; iMGC = interrogating microglia; Mt = mitochondria; N = nucleus; PcP = pericyte process; TJ/AJ = tight and adherens junction(s).

**Figure 12 medicina-59-01124-f012:**
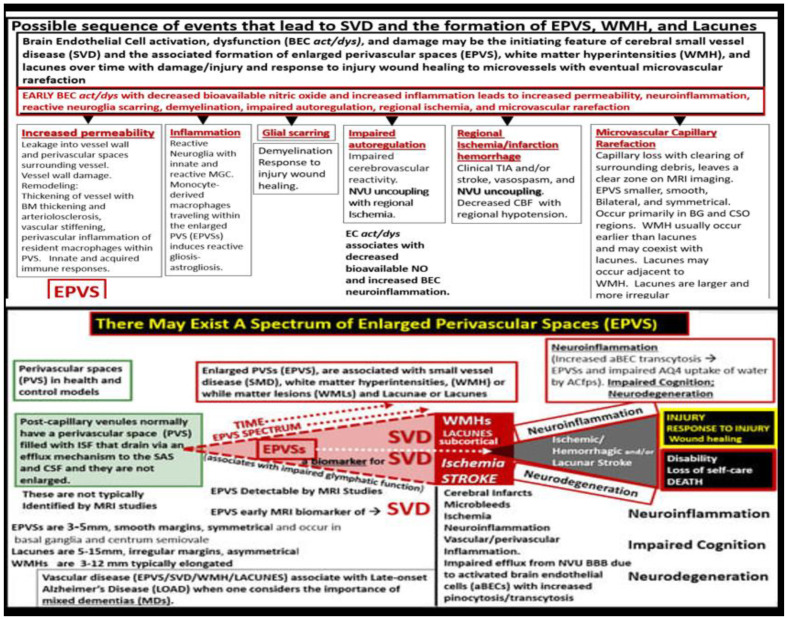
A possible sequence of events and a continuous spectrum for the development of enlarged perivascular spaces (EPVS). This modified image was provided with permission by CC 4.0 [1].

**Figure 13 medicina-59-01124-f013:**
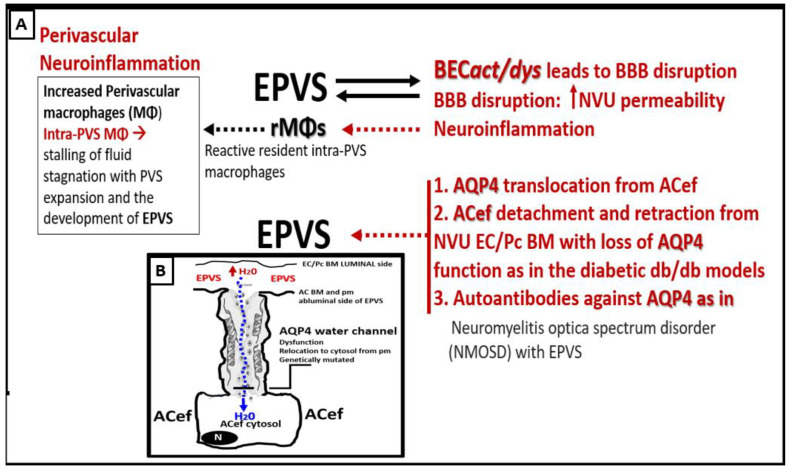
(**A**) Possible mechanisms involving the interactions between brain endothelial cell activation and dysfunction (BEC*act/dys)* and the aquaporin 4 (AQP4) water channel in the development of enlarged perivascular spaces (EPVS). (**B**) illustrates an example of the polarized AQP4 water channel at the plasma membrane (pm) of the astrocyte end-feet (ACef). BM = basement membrane; EC = brain endothelial cell; H_2_0 = water; NVU = neurovascular unit; Pc = pericyte.

## Data Availability

Data and materials will be provided upon reasonable request.

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
