# Peer review of "Brain Endothelial Cells Play a Central Role in the Development of Enlarged Perivascular Spaces in the Metabolic Syndrome"

_medicina, 2023, doi:10.3390/medicina59061124_

Round 1

Reviewer 1 Report

Thank you for the opportunity to review this manuscript entitled ‘Brain Endothelial Cells Play a Central Role in the Development of Enlarged Perivascular Spaces in the Metabolic Syndrome’ by Melvin R. Hayden. 

In the present study, authors summarized the recent advances of numerous functions of the brain capillary endothelial cells in the normal condition and in the development of the metabolic syndrome. This problem is very relevant due to the increasing of risk factors and variables associated with the Metabolic Syndrome as they age. Additionally, this also places our older society members globally at risk for accelerated brain aging,increased vulnerability to progressive neurodegeneration, impaired cognition, neurodegeneration, dementia, and age-related neurodegenerative diseases. The raise number of patients with neurodegenerative diseases worldwide and the increasing financial burden on the treatment and care of such patients increase the relevance of this study.

The manuscript is well written and logical presents review of current scientific evidence. However, some figures were recently published in Medicina 2023, 59(5), 917; https://doi.org/10.3390/medicina59050917. 

Author Response

Response to Reviewer number 1

Regarding:

Regarding manuscript ID: medicina-2421213

Brain Endothelial Cells Play a Central Role in the Development of Enlarged Perivascular Spaces in the Metabolic Syndrome

The author would first like to thank reviewer’s number 1 for taking the precious time and sharing of  knowledge in reviewing this manuscript.

Reviewer number 1 Comments and Suggestions:

Thank you for the opportunity to review this manuscript entitled ‘Brain Endothelial Cells Play a Central Role in the Development of Enlarged Perivascular Spaces in the Metabolic Syndrome’ by Melvin R. Hayden. 

In the present study, authors summarized the recent advances of numerous functions of the brain capillary endothelial cells in the normal condition and in the development of the metabolic syndrome. This problem is very relevant due to the increasing of risk factors and variables associated with the Metabolic Syndrome as they age. Additionally, this also places our older society members globally at risk for accelerated brain aging, increased vulnerability to progressive neurodegeneration, impaired cognition, neurodegeneration, dementia, and age-related neurodegenerative diseases. The raise number of patients with neurodegenerative diseases worldwide and the increasing financial burden on the treatment and care of such patients increase the relevance of this study.

The manuscript is well written and logical presents review of current scientific evidence. However, some figures were recently published in Medicina 2023, 59(5), 917; https://doi.org/10.3390/medicina59050917. 

Response to reviewer # 1:

The author wishes to thank reviewer number 1 for the kind comments.

Please note that in all figures 1, 4, 5, 6, 7, and 11 were carefully called out by utilizing permission by CC 4.0 and their respective references.

Sincerely,

Melvin R. Hayden

Submitting author 

Reviewer 2 Report

I carefully read the article titled "Brain Endothelial Cells Play a Central Role in the Development of Enlarged Perivascular Spaces in the Metabolic  Syndrome". It is a well written review paper. However, there are several issues that require revision.

- Emphasis on inflammation's role in metabolic syndrome and brain endothelial cells' involvement in inflammatory processes is advised. 

- Some of the figures are blurry (i.e. fig 11). I recommend replacing with higher quality figures.

- Conclusions are too long. This section must be the author's conclusions drawn from the text. Therefore, citations are not proper in this section. Please re-write the conclusions.

Author Response

Response to reviewer number 2

Reviewer number 2 Comments and Suggestions:

I carefully read the article titled "Brain Endothelial Cells Play a Central Role in the Development of Enlarged Perivascular Spaces in the Metabolic  Syndrome". It is a well written review paper. However, there are several issues that require revision.

- Emphasis on inflammation's role in metabolic syndrome and brain endothelial cells' involvement in inflammatory processes is advised. 

Author’s response:

Please note that on numerous occasions throughout the text that the author has stated the importance of inflammation-metainflammation and neuroinflammation.

However, in order to place more emphasis on the role of inflammation-metainflammation and neuroinflammation the author has now placed a special paragraph in the concluding paragraph of Section 2.  In lines (378-394) in blue lettering. 

- Some of the figures are blurry (i.e. fig 11). I recommend replacing with higher quality figures.

Author has now brightened and enhanced figures and especially figure 11 to provide a higher quality of images.

- Conclusions are too long. This section must be the author's conclusions drawn from the text. Therefore, citations are not proper in this section. Please re-write the conclusions.

Author has now removed all references from the conclusion section and wishes to thank the reviewer for these suggestions and recommendations.  However, author did retain the following regarding IPAD in the arterioles and precapillary segments of the  IPAD statement by Carare et al, with reference now [58] at the concluding paragraph of Section 3. (lines 573-577 in blue lettering) and also, renumbered the references in the text and reference sections accordingly.

Reviewer # 2 comments and suggestions were very helpful in making this paper better for our readers and I really appreciated these comments and suggestions.

Author is now hopeful that the revised manuscript will now be found suitable for publication in the Medicina Journal.

Sincerely,

Melvin R. Hayden

Submitting author  

Round 2

Reviewer 1 Report

I withdraw my remarks if such presentation of figures does not contradict the editorial policy of the journal